# Biological Transformation of Zearalenone by Some Bacterial Isolates Associated with Ruminant and Food Samples

**DOI:** 10.3390/toxins13100712

**Published:** 2021-10-09

**Authors:** Sharif Zada, Sadia Alam, Samha Al Ayoubi, Qismat Shakeela, Sobia Nisa, Zeeshan Niaz, Ibrar Khan, Waqas Ahmed, Yamin Bibi, Shehzad Ahmed, Abdul Qayyum

**Affiliations:** 1Department of Microbiology, Hazara University, Mansehra 21120, Pakistan; Sharifkhan847@gmail.com (S.Z.); themicrobiologist@gmail.com (Z.N.); shehzadlughmani@gmail.com (S.A.); 2Department of Microbiology, The University of Haripur, Haripur 22620, Pakistan; sobia@uoh.edu.pk (S.N.); waqas.ahmed@uoh.edu.pk (W.A.); 3College of Humanities and Sciences, Prince Sultan University, Riyadh 11586, Saudi Arabia; sayoubi@psu.edu.sa; 4Department of Microbiology, Abbottabad University of Science and Technology, Havelian 22060, Pakistan; qismatshakeela@yahoo.com (Q.S.); abrar@aust.edu.pk (I.K.); 5Department of Botany, PMAS-Arid Agriculture University Rawalpindi, Rawalpindi 46300, Pakistan; dryaminbibi@uaar.edu.pk; 6Department of Agronomy, The University of Haripur, Haripur 22620, Pakistan

**Keywords:** zearalenone, fusarium, ruminants, ELISA, HPLC, 16SrRNA

## Abstract

Zearalenone (ZEA) is a secondary metabolite produced by *Fusarium* spp., the filamentous fungi. Food and feed contamination with zearalenone has adverse effects on health and economy. ZEA degradation through microorganisms is providing a promising preventive measure. The current study includes isolation of 47 bacterial strains from 100 different food and rumen samples. Seventeen isolates showed maximum activity of ZEA reduction. A bacterial isolate, RS-5, reduced ZEA concentration up to 78.3% through ELISA analysis and 74.3% as determined through HPLC. Ten of the most efficient strains were further selected for comparison of their biodegradation activity in different conditions such as incubation period, and different growth media. The samples were analyzed after 24 h, 48 h, and 72 h of incubation. De Man Rogosa Sharp (MRS) broth, Tryptic soy broth, and nutrient broth were used as different carbon sources for comparison of activity through ELISA. The mean degradation % ± SD through ELISA and HPLC were 70.77% ± 3.935 and 69.11% ± 2.768, respectively. Optimum reducing activity was detected at 72 h of incubation, and MRS broth is a suitable medium. Phylogenetic analysis based on 16S rRNA gene nucleotide sequences confirmed that one of the bacterial isolate RS-5 bacterial isolates with higher mycotoxin degradation is identified as *Bacillus subtilis* isolated from rumen sample. B05 (FSL-8) bacterial isolate of yogurt belongs to the genus *Lactobacillus* with 99.66% similarity with *Lactobacillus delbrukii*. Similarly, three other bacterial isolates, D05, H05 and F04 (FS-17, FSL-2 and FS-20), were found to be the sub-species/strains *Pseudomonas gessardii* of genus *Pseudomonas* based on their similarity level of (99.2%, 96% and 96.88%) and positioning in the phylogenetic tree. Promising detoxification results were revealed through GC-MS analysis of RS-5 and FSL-8 activity.

## 1. Introduction

Zearalenone (ZEA) is a non-steroidal estrogen synthesized biologically by *Fusarium* spp., including *F. cerealis, F. graminearum, F. culmorum, F. semitectum* and *F. equiseti* through a polyketide pathway. ZEA is placed by international the Agency for Research on Cancer in group 3 carcinogens [1]. Mycotoxins occurrence in food and feed is regulated now by EU legislations [2,3]. ZEA is a much stabler compound during both processing/cooking and the storage/milling of food. Its presence is detected in various grain products, such as beers, bread and some processed feeds [4,5].

ZEA has many adverse effects on the living body. It may alter the physiology of reproductive system of animals [1,5] and is also related with precocious puberty of pre-pubertal girls [6,7,8,9]. Genotoxic and hematotoxic effects are also reported apart from its estrogenic properties [5,10]. The estrogenic effects of ZEA include reduced sperm counts and serum testosterone levels, infertility, incidence of pregnancy and change in the level of progesterone [11].

ZEA is absorbed rapidly and metabolized initially by tissues of intestine and hepatocytes. These tissues initiate biotransformation of ZEA into α- and β-zearalenol (α- and β-ZOL) that are major biologically active reduced metabolites. The estrogenic ability of ZEA is mediated by their binding to estrogen receptors (ER), and they are as potent as genistein and coumestrol, two endocrine-disrupting phytoestrogens [5]. The most frequently recorded ZEA derivatives are α- and β-ZOL [12,13,14].

Several physical and chemical technologies used for inactivation or elimination of ZEA have been reported [15]. The use of chemical fungicides is used as main solution to control the growth and survival of fungi in crops, but this method has a negative effect on the environment and causes pollution. After their application, they have several negative implications, e.g., observed and induced resistance to introduced chemical substances that reduce growth of pathogen [16]. Microorganisms exhibit a viable alternative to biodegrade mycotoxins including ZEA.

Biotransformation is considered as a cost effective and environmentally friendly solution for removing ZEA, without damaging the feed nutrition [17]. Previously, some ZEA-degrading bacterial strains and key enzymes related to the ZEA neutralization were investigated. ZEA could be converted to a much less estrogenic product by *Clonostachys rosea*; furthermore, the hydrolase associated with the detoxification has been purified to homogeneity and its gene (zhd101) has been isolated [18,19]. Binding kinetics of *Lactococcus* and *Bifidobacterium* and subsequent spectroscopic studies revealed non-homogenous and homogenous biosorption of zearalenone, respectively [20,21]. Microorganisms in stress conditions compel to maximize their cellular sorption ability of ZEA [21].

Mokoena et al. [22] obtained the potential of fermentation of lactic acid bacteria (LAB) in neutralization of a 56% to 67% decrease in the third and fourth day of incubation, respectively. *Lactobacilli* and *Bacilli* are also able to bind and adsorb ZEA [23]. 

Lei et al. [24] have isolated a bacterial strain of *Bacillus subtilis* ANSB01G from broiler intestinal chyme that showed ability to remove 88.65% of ZEA. Moreover, Wang et al. [25] have selected a ZEA-degrading bacterium *Bacillus pumilus* ES-21 by esterase activity assay.

The main objective of the current research work was to isolate and explore bacterial strains from different food and rumen samples that have the ability to neutralize ZEA into non-hazardous compounds.

## 2. Results

Total of 47 bacterial strains were isolated from 100 different food and rumen samples. Seventeen of them showed high ZEA reducing activity to varying degrees. 

### 2.1. Gram’s Staining and Morphological Characterization

Among 47 bacterial isolates, 19 were Gram positive cocci, 1 was Gram negative cocci, 25 were Gram positive bacilli and 2 were Gram negative bacilli. Seventeen selected isolates exhibited are presented in Table 1. Cell morphology showed that RS-6 had short rods. Most of the isolated bacterial colonies were small, circular shape with shiny whitish cream or brownish colored with convex evaluation. Some colonies were large irregular, non-shiny having rough surface. The isolated colonies were examined under light compound microscope after Gram staining. The bacterial isolate RS-5 that showed maximum reducing activity of ZEA was found in irregular colonial form while FSL-8 formed circular shiny colonies on growth medium. These bacteria were observed rod shape under microscope (Figure 1). Most of the isolated bacteria in circular colonial form appeared to be *Lactobacillus* spp. and were seen short rods under light compound microscope. 

### 2.2. Biochemical Identification

Biochemical tests were performed for identification of all the 47 strains (coded as FSL-1 to FSL-30 and RS-1 to RS-17). All these biochemical analyses helped to differentiate and identify bacteria on the basis of their enzymes production. Out of 47 bacterial strains, 32 were found catalase positive, 33 oxidase positive, 33 amylase positive, 9 indole positive and 19 were found positive to ferment carbohydrates. Dextrose and sucrose were used as a carbohydrate source for carbohydrate fermentation tests (Table 1). 

The bacterial isolate RS-5 was catalase positive, oxidase positive, indole negative and carbohydrate fermentation test was also negative, presumably identified as *B. subtilis* while FSL-8 was catalase negative, oxidase negative, indole negative and carbohydrate fermentation test was positive and identified as *Lactobacillus delbrueckii.* The isolated bacteria coded as FSL-2, FSL-17 and FSL-20 were Gram negative rods and identified as *Pseudomonas gessardii* by performing 16SrRNA sequencing. The Gram positive rod shape bacteria in cluster form isolated from rumen sample of ruminant also possess high degradation activity of ZEA and identified as *Bacillus subtilis* by molecular analysis. 

### 2.3. Molecular Identification

The phylogenetic analysis confirms the molecular identification of 4 different samples based on constructing phylogenetic tree with most of the closest member of bacterial candidates (Figure 2, Figure 3 and Figure 4).

The nucleotide sequence similarity of sample B06 (FSL-8) was found to be 96.66% with the bacterial species of *Lactobacillus delbrueckii* isolated from yogurt. Due to partial amplification and limited size of the DNA fragment it is presumed to be sub-species or strain of *Lactobacillus delbrueckii*, despite of low similarity level but close position in phylogenetic tree in cluster adjacent to the similar bacterial strain (Figure 2). This sequence was submitted in NCBI with accession no. MZ452409.

When the nucleotides sequences of bacterial isolates D05 (FS-17), H05 (FSL-2) and F04 (FS-20), were blasted in NCBI, Ezbiocloude Taxon showed sequence similarity (99.2%, 96% and 96.88%) with *Pseudomonas gessardii* obtained from onion bulb in Taiwan. The partial sequence size and variable similarity index of all three bacterial isolates indicated that, each of the bacterial isolate belongs to Genus *Pseudomonas*. The similarity percentage of H05 (Accession No MZ452411) represents *Pseudomonas gessardii* while D05 (Accession No. MZ452412), and F04 (Accession No. MZ452410) represents the same genus, i.e., *Pseudomonas*, however with different strains number due to sequences variation with each other and their position on phylogenetic tree within the cluster having *Pseudomonas gessardii*. As the partial sequence data for D05 and F04 is less than 1300 bp nucleotide, these two will be considered as *Pseudomonas* genus. The clear position of sample strains with describe strains of *Pseudomonas* in the phylogenetic tree with significant bootstrap value further validates the confidence of the tree, and thus confirms the described classification as above (Figure 3).

Similarly, a nucleotide sequence of isolated bacteria from rumen sample RS-5 was also selected for amplification with 16SrRNA bacterial universal prime 27F1492R. Ezbiocloude taxon showed sequence similarity (99%) with *Bacillus subtilis.* The result obtained from phylogenetic tree reveals that the isolated bacteria possess similarity with *B. subtilis* (Figure 4).

### 2.4. Screening of Zearalenone Biodegrading Bacterial Strains through ELISA 

ELISA results depicted that 17 bacterial strains from different food sources and rumen samples showed maximum ZEA neutralization activity in MRS broth at 37 °C after 72 h of incubation. There was no significant variation in different media, used for ZEA bio-degradation. 

#### Detection of Zearalenone through ELISA

The maximum biodegradation activity was exhibited by bacterial strain *Bacillus subtilis* RS-5 (78.3%), *Lactobacillus delbrueckii* FSL-8 (78%) activity while *Pseudomonas gessardii* (FSL-2) showed (72.8%) biodegradation activity after 72 h of incubation in MRS broth (Figure 2, Figure 3, Figure 4 and Figure 5).

Absorbance percentage of efficient bacterial samples, degraded amount of ZEA and log concentration of different bacterial strains used in neutralization of ZEA was calculated from ELISA results. Each bacterial strain has different potential to degrade ZEA in MRS broth on 37 °C (Table 2).

Ten of the most efficient isolates were further selected for comparison of their ZEA detoxification ability. The biotransformation activity of isolated bacteria checked after incubation of 24 h, 48 h and 72 h revealed promising results of ZEA biodegradation. RS-5 reduced 78.3% ZEA in nutrient broth after 72 h of incubation. FSL-8 degraded 78% ZEA in MRS broth after 72 h (Table 3).

The average percentage biodegradation of ZEA was compared after different incubation times and different carbon sources used (*n* = 3). The comparison analysis showed that maximum reduction of ZEA was reported after 72 h of incubation in MRS broth by bacterial isolate FSL-8. The average biodegradation percentage of RS-5 in MRS broth after 72 h of incubation was 78.3% while of FSL-8 was 78% (Table 3). It was noted that MRS broth was most suitable medium for biodegrading probiotic isolates and maximum activity incubation duration was 72 h (Figure 6).

### 2.5. Analysis of Bio-Transforming Ability of Bacterial Isolates through HPLC

HPLC analysis revealed that 17 probiotic isolates possess ZEA neutralization ability (Table 4). These isolates neutralized ZEA up to various ranges. RS-5 exhibited maximum neutralizing activity with 74.3% activity followed by FSL-8 (72.4%). RS-5 was isolated from the rumen of buffalo, while FSL-8 was from yogurt sample. The lowest neutralizing activity among these 17 strains was found by the strain coded as RS-13 that show 64.4% activity. Other strains showed ZEA reduction activity between the highest value 74.3% and lowest 64.4%. ZEA standard peak was obtained at retention time 4.498 min.

The area of HPLC peak of the samples was compared with the area of the ZEA standard (control sample). The peak obtained in FSL-8 on retention time (RT) 4.361 is the peak of zearalenone which is compared with the same peak of control sample on retention time 4.498 (Figure 7). The height of ZEA peak of sample RS-5 was 8363 and of control sample was 30,274 (Figure 8).

The comparison of biodegradation percentage of different bacterial isolates showed variable biodegrading ability through HPLC (Figure 9).

### 2.6. *t*-Test:/Statistical Analysis

*t*-test was applied to find the probability value (*p* value) of results obtained from ELISA and HPLC. The Mean ± SD of ELISA and HPLC was 70.77 ± 3.935 and HPLC was 69.11 ± 2.768, respectively. The probability value (*p* value) obtained through *t*-test was *p* < 0.05 by performing two tailed *t*-test which is significant. The Mean ± SD obtained from ELISA and HPLC indicated that results do not vary significantly (Table 4).

### 2.7. GCMS Analysis

GCMS analysis indicated absence of peak for zearalenone after treatment with isolated bacterial strain (RS-5). Some secondary metabolites and some proposed bio-transformed products of zearalenone were observed after retention time of 0.41, 2.69, 5.61 and 7.71, respectively (Figure 10). The most prominent compounds detected were 1 propene,3,3ꞌ-oxy bis, pyrrolo [1.2-q] pyrazine-1,4-dione,hexahydro, Hexasiloxane,1,1,3,3,5,5,7,7,9,9,11,11 dodecamethyl and Octasiloxane,1,1,3,3,5,5,7,7,9,9,11,11,13,13,15,15-hexadecamethyl-(Table 5). There was no evidence of α zearalenol and β zearalenol in the mixture of degraded compounds and bacterial metabolites (Table 5).

## 3. Discussion

Biodegradation is a universal strategy used for the management of mycotoxins; it is considered one of the most specific, efficient, and environmentally friendly methods in reducing or eliminating the possible mycotoxins from food and feed. In the current research study, 17 strains showed reducing ability of zearalenone in different ranges. The molecular evidence based on 16S rRNA gene sequences reveals that RS-5 with significant mycotoxin degradation was *Bacillus subtilis* while FSL-8 (B06) was the sub-species or strain of *Lactobacillus delbrueckii*. It was confirmed by data obtained through NCBI, Ezbiocloude taxon and its position on a phylogenetic tree constructed with most of the species of *Lactobacilli*. Similarly, when three other bacterial isolates, i.e., FSL-2, FS-17 and FS-20 (HO5, DO5 and F04, respectively), were studied based on 16S rRNA gene nucleotide sequences, it was found that all three bacterial isolates belong to *Pseudomonas* genera and its species *P. gessardii*. The data retrieved through NCBI and their positioning on phylogenetic tree confirmed that all the three bacterial isolates might be sub-species or strains of *P. gessardii*. The maximum overall biotransformation activity was reported of RS-5 (74.3%) followed by FSL-8 (72.4%) after 72 h of incubation through HPLC and ELISA result was 78.3% for RS-5 and 78% for FSL-8 (MRS broth). Bacterial isolate coded as RS-5 was identified as *B. subtilis* and FSL-8 was *Lactobacillus delbrueckii* through biochemical tests, microscopy and molecular study. Different media MRS agar, tryptic casein soy broth and nutrient broth used for ten more efficient strains out of 17 strains for comparison of better activity in different conditions through ELISA showed non-significant difference of ZEA neutralization. This finding may be due to the fact that probiotic isolates showed promising growth in MRS broth as compared to nutrient and tryptic casein broth. The maximum neutralization result was obtained after 72 h of incubation in MRS broth. It is also a known fact that increases in incubation time increase bacterial viability. These results are comparable to findings of previous scientists [26,27]. Proteins and peptide of culture medium may affect ZEA concentration as these adsorb the toxin on their surface [27]. Our findings suggest the contrary that ZEA must be present in GC-MS analyzed results if adsorbed on molecules surface. Various fungi, bacteria and yeasts can convert ZEN to α- and β-ZOL, but both of these are also toxic [28]. Our findings indicate that ZEA was not transformed in α- and β-ZOL. The compounds detected through GC-MS were reported as nontoxic as compared to these two compounds. The mean degradation ability percentage through ELISA and HPLC was determined as 70.77% and 69.11%, respectively. Two tailed *t*-test revealed a significant difference between both methods, but proved high specificity of ELISA technique as compared to HPLC.

Lei et al. [24] analyzed the biotransformation activity of *B. subtilis* ANSB01G from normal broiler intestinal chyme and found 88.6% ZEA degradation. Cho et al. [29] also identified *B. subtilis*, as potential ZEA degrading bacterium that could biodegraded more than 95% of the ZEA in lysogeny broth in 24 h. Tiemann et al. [13] also observed that *Lactobacillus* possess ZEA degrading ability up to some extent.

Mokoena et al. [22] obtained the potential of fermentation of lactic acid bacteria (LAB) in degradation of a 56% to 67% decrease in the third and fourth day of incubation, respectively. Lactobacilli and Bacilli are also able to bind and adsorb ZEA [23].

Zearalenone is chemically (4*S*,12*E*)-16,18-dihydroxy-4-methyl-3-oxabicyclo[12.4.0] octadeca-1(14),12,15,17-tetraene-2,8-dione (National Center for Biotechnology Information). GCMS analysis of degraded zearalenone reveal that there is a possibility that 9-Octadecenoic acid, 1,2,3-propanetriyl ester, (E,E,E)- and Pyrrolo [1,2-a] Pyrazine,-1,4-dione, hexahydo-3-(2-methylpropyl) are bio-transformed products of zearalenone. This conversion is possible by combination of bio-products of *Bacillus subtilis* and zearalenone. All probable transformed products in present research, analyzed through GCMS are potentially non-toxic or less toxic as revealed by previous studies [30,31]. Vanhoutte et al. [32] also reported the production of ether and ester derivatives by *Bacillus subtilis*. Furthermore these derivatives are considered to have important applications as they are less toxic to human and environment [23,33]. Present findings have explored in vitro degradation of ZEA through probiotic isolates and indicate their future potential use of FSL-8 and RS-5 in biotransformation of zearalenone in food and feed. Labeled isotope mass spectrometry technique may be used to assess the exact transformed product by FSL-8 and RS-5.

## 4. Conclusions

From current research study, it is concluded that probiotics isolated from different food sources and rumen of ruminants exhibited promising abilities to neutralize zearalenone. The mechanism involved in ZEA detoxification through probiotic isolates can be further explored through liquid chromatography and labeled carbon mass spectrometry can be utilized for confirmation of transformed products. These neutralizing isolates can be applied with or as food/feed additive to combat ZEA intoxication.

## 5. Materials and Methods

### 5.1. Chemicals and Reagents Used in Current Research Study

Zearalenone standard was purchased from Sigma-Aldrich Co., St. Louis, MO, USA, CAS No: 17924-92-4. Elabscience ZEA (Zearalenone) ELISA Kit Catalog No: E-TO-E002 96T (St. Louise, MO, USA) was used for quantitative determination of ZEA in samples. All the chemicals used in analysis of biotransformation of ZEA were analytical reagent grade.

### 5.2. Stock and Working Solution Preparation for ELISA and HPLC

A stock solution of ZEA was prepared by dissolving the solid standards of ZEA (Sigma-Aldrich Co., St. Louis, MO, USA) in acetonitrile (5 mg mL^−1^) and stored at −20 °C until use. Working solution (25 ng mL^−1^) from stock solution was prepared in 1% dimethyl sulfoxide (DMSO) for enzyme-linked immunosorbent assay (ELISA) and 32 µg mL^−1^ in concentrated 99.9% HPLC grade methanol (Sigma-Aldrich, St. Louise, MO, USA) for high performance liquid chromatography (HPLC).

### 5.3. Collection of Samples and Isolation of Bacteria

The dairy products, fermented foods and rumen samples were collected randomly from dairy shops, pickle shops and slaughterhouses at Haripur, Khyber Pakhtunkhwa, Pakistan and were stored in refrigerator at 4 °C until further analysis. The food samples were collected in clean shopping bags, while rumen samples were collected from fresh rumen through transport swabs. Code FSL was used for bacterial strains isolated from different foods, while RS code was used for bacterial isolates of rumen samples. Serial dilution method was used for bacterial isolation from different samples and each dilution was spread over the de Man, Rogosa, and Sharpe agar (MRS agar) of Sigma-Aldrich, St.Loise, MI, USA and incubated for 24 h at 37 °C [34].

### 5.4. Microbial Identification

Morphological examination of isolated bacteria was done by observing morphological characteristics of each colony of all the isolated bacteria [35]. Cultural and biochemical characterization of all bacterial isolates were carried out by Gram staining, catalase test, oxidase test, amylase test, indole test and sugar fermentation test. Dextrose and sucrose (Sigma-Aldrich, St.Loise, USA) were used for sugar fermentation test [35,36].

#### Molecular Identification

DNA of the most potent bacterial isolates was extracted through phenol-chloroform method and sequenced through Sanger Sequencing technique for 16S rRNA gene nucleotide sequences. The bacterial nucleotides were sent to Macrogen, Seoul, Korea for sequencing. Universal primer 27F 1492R was used for bacterial strains identification. After retrieving nucleotides, sequences were blasted in NCBI and Ezbiocloude taxon to confirm their identity. Phylogenetic analysis was conducted by using the latest version of MEGA-X software. The bacterial isolates with highest mycotoxin neutralization activates were selected for amplification with 16S rRNA bacterial.

### 5.5. Analysis of ZEA Reduction Ability of Isolated Bacteria through ELISA

Working solution (25 ng mL^−1^) of ZEA was prepared from stock solution (5 mg mL^−1^) for ELISA analysis. To make working solutions, ZEA was dissolved in 1% DMSO and then mixed in MRS broth in Eppendorf tubes to get concentration of 25 ng mL^−1^. Overnight fresh bacterial culture was mixed in each tube and then incubated on 37 °C for 72 h. After performing ELISA of all isolates, 17 potent strains were obtained in which 10 most efficient strains were further selected for comparison of their activity in different conditions for further ELISA analysis. Different incubation times such as 24 h, 48 h and 72 h were selected for comparison of their activity. Similarly, different culture media such as MRS broth, tryptic soy broth (TSB) and nutrient broth were used as different carbon source for comparison. Degraded ZEA was quantified by comparing samples with ZEA standards of different concentrations available in Elabscience ZEA (zearalenone) ELISA Kit (0 ng mL^−1^, 0.3 ng mL^−1^, 0.9 ng mL^−1^, 2.7 ng mL^−1^, 8.1 ng mL^−1^, 24.3 ng mL^−1^).

ELISA Reader model # BioTek EL ×800 was used to determine the absorbance values of different samples at 450 nm wavelength.

### 5.6. Analysis of ZEA Reduction Ability of Bacterial Isolates through HPLC

HPLC was performed for testing of ZEA reducing activity of isolated bacteria. First, 10 mL Lysogeny broth (LB broth) containing 32 μg mL^−1^ of ZEA was inoculated with 100 µL of fresh bacterial culture. The control sample was also prepared in the same way without inoculation of bacterial culture. After 72 h of incubation, the bacteria were sub-cultured by adding 100 µL of the fresh bacterial culture to 10 mL LB broth containing 32 μg mL^−1^ of ZEA and were properly incubated in shaking incubator at 37 °C with 200 rpm for 72 h. Sub-culturing was repeated for three times to enrich ZEA degrading bacteria. At the end of each sub-culturing, 1 mL culture was mixed with 3 mL methanol and then vortex well for 1 min. Then, 1 mL mixture was then centrifuged at 12,000 rpm for 10 min. The supernatants containing residual ZEA was determined by performing HPLC.

#### HPLC Conditions

The concentration of ZEA in food and rumen samples were determined by HPLC (Weather Folds Pharmaceuticals Hattar industrial Estate Hattar Khyber Pakhtunkhwa Pakistan) equipped with a Waters 600 HPLC System with waters 996 photodiode array detector, and an Agilent Zorbax Eclipse XDB-C18 column (250 × 4.6 mm i.d., 5 μm, Agilent, Santa Clara, CA, USA). The mobile phase was prepared as acetonitrile: methanol: water (46:8:46, *v/v/v*), which has been filtered properly through a membrane (0.45 μm) and then degassed in ultrasonic for 10 min before HPLC. The flow rate of mobile phase was 1 mL/min with 10 μL injection volume. The concentration of ZEA was determined at 274 nm (excitation) and 460 nm (emission), and the retention time was 10 min. The reduction rate of ZEA was calculated as:Zr = (1 − Ct/Cck) × 100%(1)where Zr represents the ZEA-Reduction (%), Cck and Ct were the concentration of ZEA in control samples and experimental treatments, respectively. The control sample and the experimental treatments had the same initial concentration of ZEA.

### 5.7. Gas Chromatography-Mass Spectrometry

#### 5.7.1. Sample Preparation and Zearalenone Extraction

Sample was prepared by modifying Mirocha et al. method [37]. Zearalenone (ZEA) working solution (32 µg mL^−1^) in acetonitrile was prepared for zearalenone biodegradation through GC-MS technique. The ZEA sample was treated with fresh bacterial cultures and incubated for 72 h at 37 °C. Working solution (500 µL) and 500 µL of nutrient broth having microbial culture were put in Eppendorf tubes and were kept in shaking incubator for 72 h at 37 °C. After 72 h, the sample was centrifuged at 12,000 rpm.

The culture supernatant containing ZEA (1.5 mL) was taken and acetonitrile+water was added in 86:14 *v/v* ratio. The mixture was kept for 24 h after capping. The extract (1.5 mL) was passed through a minicolumn syringe containing C-18:aluminium oxide in 1:3 *w/w* ratio. The elute (1 mL) was evaporated and dried after passing through dram tube for 1 h with nitrogen (Sigma-Aldrich, St. Louis, MO, USA). Then, 20 µL TMS reagent was added to the dried toxin. The vial was shaken, and the reaction was allowed to take place for 15 min. Isooctane 180 µL was added and then 200 µL sterilized distilled water was added to the reaction mixture. The mixture was vortexed to get a clear layer of isooctane. The transparent layer from above was transferred to a GC vial for analysis through GC-MS instrument “Thermo Scientific” (GC-MS) DSQ II Waltham, MA USA 02451.

#### 5.7.2. GCMS Condition

The GC was equipped with a TR-5M5 capillary column of 30 m in length, film thickness 0.25 µm and internal diameter of 0.25 mm. Helium was used as the carrier gas. Then, 1 µL volume of each sample was injected to inlet by auto sampler. Uninoculated ZEA working solution in nutrient broth served as control.

The initial column temperature was held at 150 °C for 1 min and then ramped to 280 °C at a rate of 30 °C min^−1^ and held constant for 3 min. This was followed by post run of 325 °C for 2.5 min [27].

## Figures and Tables

**Figure 1 toxins-13-00712-f001:**
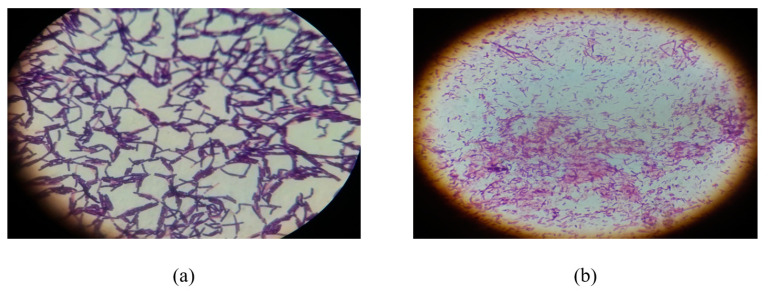
Microscopic view (100×) of (**a**) *Bacillus subtilis* isolate of buffalo rumen (**b**) *Lactobacillus delbrueckii* isolate of yogurt sample that show maximum biodegrading ability of ZEA.

**Figure 2 toxins-13-00712-f002:**
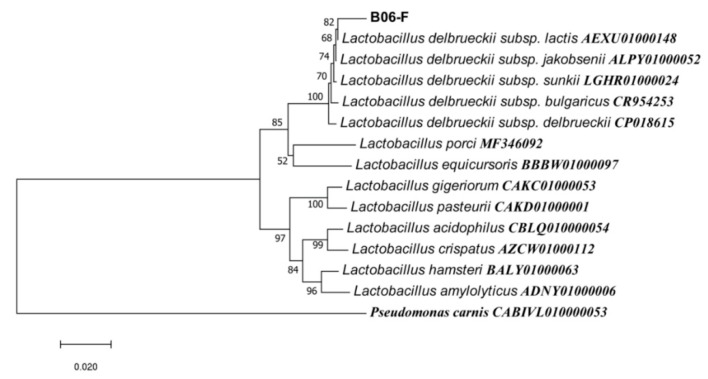
The phylogenetic tree based on 16S rRNA gene of mycotoxin degrading *Lactobacillus* strain through Neighbor-Joining method. Kimura 2-parameter model with bootstrap value (*n* = 100) was used for computing evolutionary distances. Most of the species of Lactobacilli with close resemblance to the sample were used along with *Pseudomonas carnis* CABIVL010000053 was used as an out-group. All positions with <95% site coverage was eliminated.

**Figure 3 toxins-13-00712-f003:**
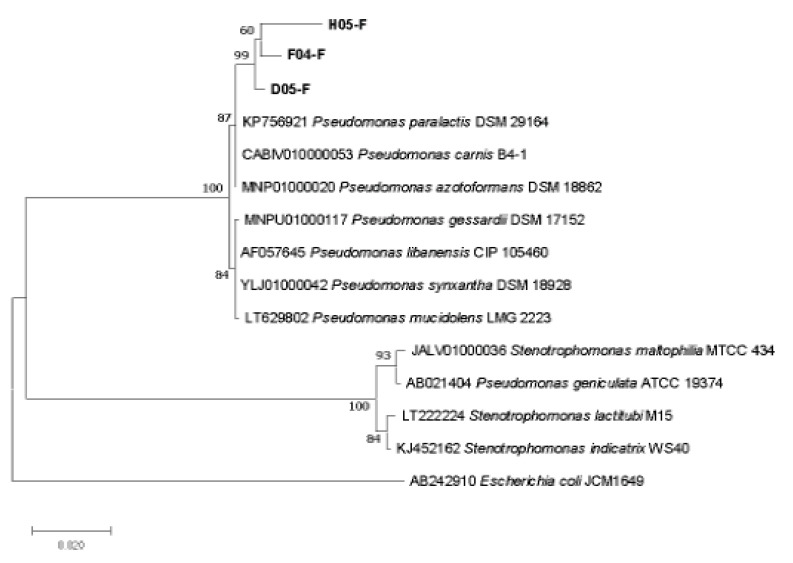
The phylogenetic tree based on 16S rRNA gene of mycotoxin degrading *Pseudomonas* strain through Neighbor-Joining method. Kimura 2-parameter model with bootstrap value (*n* = 100) was used for computing evolutionary distances. Most of the species of *Pseudomonas* with close resemblance to the sample were used along with *Escherichia coli* AB242910 was used as an out-group. All positions with <95% site coverage were eliminated.

**Figure 4 toxins-13-00712-f004:**
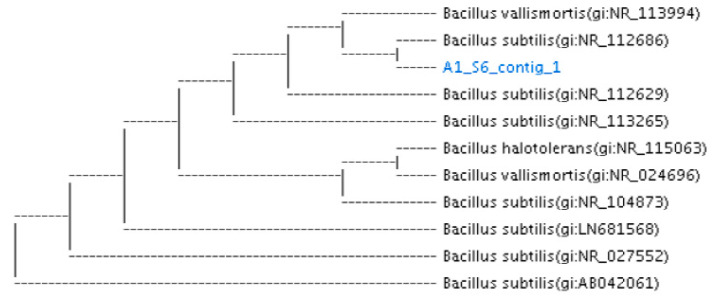
The phylogenetic tree based on 16S rRNA gene of mycotoxin degrading *Bacillus subtilis* strain through Neighbor-Joining method. Kimura 2-parameter model with bootstrap value (*n* = 100) was used for computing evolutionary distances. RS-5 harbored 99% similarity with *Bacillus subtilis*.

**Figure 5 toxins-13-00712-f005:**
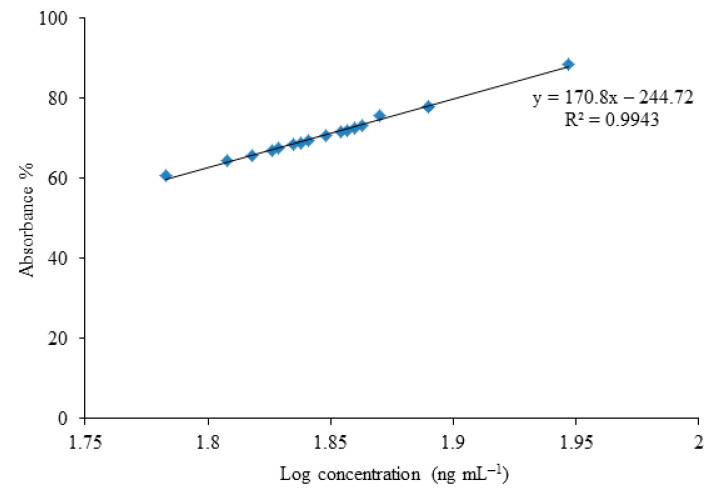
Log concentration and absorbance percentage of zearalenone through ELISA of potent bacterial strains show maximum biotransformation activity.

**Figure 6 toxins-13-00712-f006:**
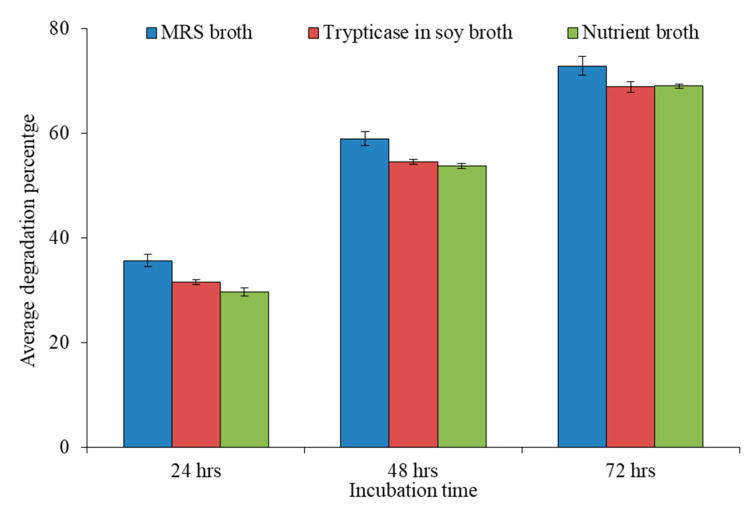
Comparison of average biodegradation percentage of ZEA in different culture media at different incubation time.

**Figure 7 toxins-13-00712-f007:**
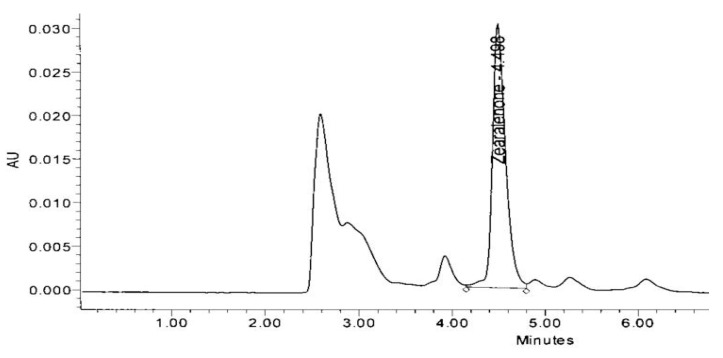
High performance liquid chromatogram of ZEA standard.

**Figure 8 toxins-13-00712-f008:**
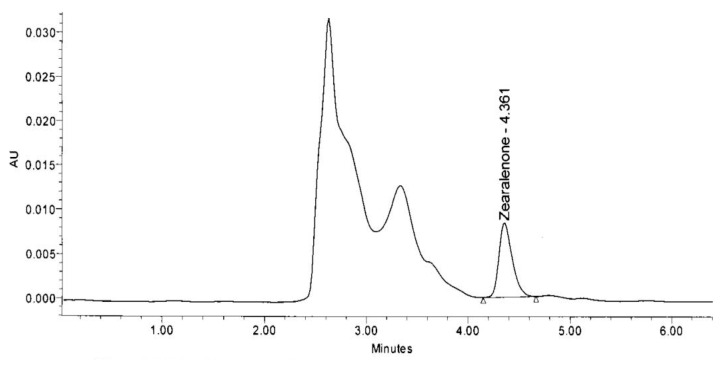
HPLC result of zearalenone degrading ability of RS-5 bacterial isolate obtained from yogurt sample.

**Figure 9 toxins-13-00712-f009:**
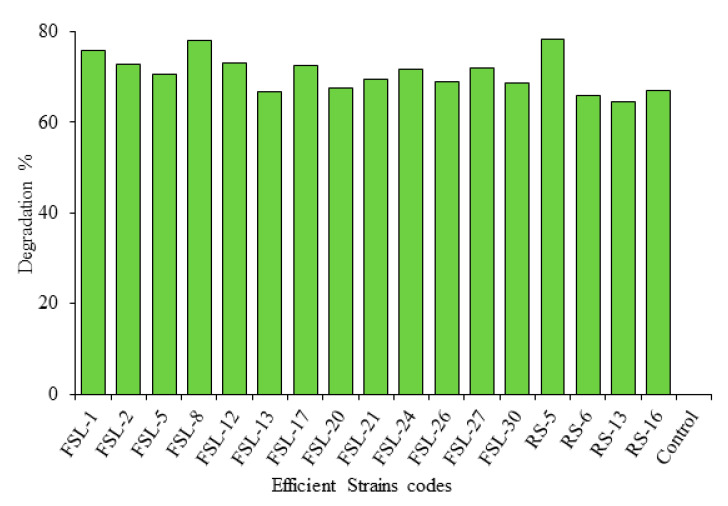
Comparison of percentage degrading of ZEA by potent bacterial strains through HPLC.

**Figure 10 toxins-13-00712-f010:**
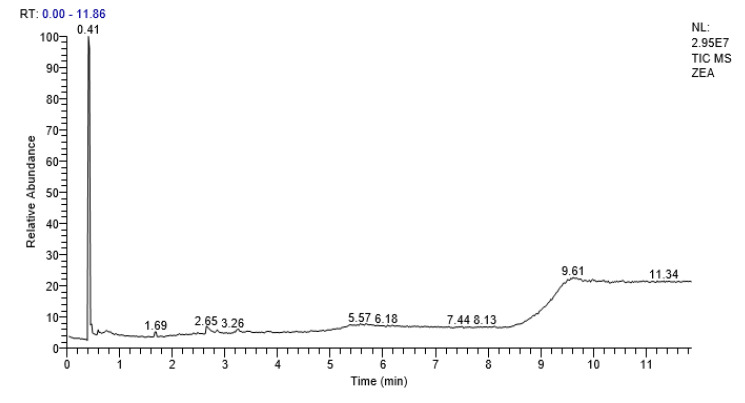
GC-MS chromatogram of ZEA after 72 h incubation of RS-5.

**Table 1 toxins-13-00712-t001:** Biochemical tests and Gram staining result of efficient bacterial strains that show maximum activity of ZEA biodegradation.

S.No	Strain Code	Catalase Test	Oxidase Test	Amylase Test	Indole Test	Carbohydrate Fermentation Test	Gram Staining
1	FSL-1	–	–	+	–	+	Gram positive rods
2	FSL-2	+	+	+	–	+	Gram negative rods
3	FSL-5	+	+	–	–	+	Gram Positive short rods
4	FSL-8	–	–	–	–	+	Gram positve rods
5	FSL-12	–	–	+	–	+	Gram positive rods
6	FSL-13	+	+	–	–	–	Gram positive short rods
7	FSL-17	+	+	+	–	+	Gram negative rods
8	FSL-18	–	–	+	–	+	Gram positive rods
9	FSL-20	+	+	+	–	+	Gram negative rods
10	FSL-21	+	+	–	–	–	Gram positive short rods
11	FSL-24	–	–	+	–	+	Gram positive rods
12	FSL-26	–	–	+	–	+	Gram negative rods
13	FSL-27	–	–	+	–	+	Gram positive rods
14	RS-5	+	+	+	–	+	Gram positive rods
15	RS-6	+	+	–	–	–	Gram Positive short rods
16	RS-13	+	+	–	–	+	Gram positive rods
17	RS-16	+	+	+	–	+	Gram positive rods

**Table 2 toxins-13-00712-t002:** Comparison of absorbance percentage and various concentrations of ZEA in potent bacterial strain samples.

S.No	Strain Code	Absorbance Percentage (*Y*-Axis)	Log Concentration (*X*-Axis)	Initial Concentration (ng mL^−1^)	Final Concentration (ng mL^−1^)	Degraded Amount (ng mL^−1^)
1	FSL-1	75.7	1.87	25	6.07	18.93
2	FSL-2	72.8	1.86	25	5.55	19.45
3	FSL-5	70.6	1.848	25	7.35	17.65
4	FSL-8	78	1.89	25	5.5	19.50
5	FSL-12	73.1	1.863	25	6.72	18.28
6	FSL-13	66.8	1.81	25	8.9	16.10
7	FSL-17	72.6	1.86	25	6.85	18.15
8	FSL-20	67.5	1.829	25	8.1	19.50
9	FSL-21	69.5	1.841	25	7.62	17.38
10	FSL-24	71.6	1.854	25	7.22	17.78
11	FSL-26	69	1.838	25	7.75	17.25
12	FSL-27	72	1.857	25	7.0	18.00
13	FSL-30	68.5	1.835	25	7.8	17.20
14	RS-5	78.3	1.89	25	2.9	22.10
15	RS-6	65.9	1.818	25	8.75	16.25
16	RS-13	64.4	1.808	25	8.9	16.10
17	RS-16	67	1.826	25	8.25	16.75
18	Control	0	1.39	25	25	0

**Table 3 toxins-13-00712-t003:** Biodegradation percentage of ZEA through efficient bacterial strains in different culture media at 37 °C.

S.No	Strain Code	Degradation % 24 h			Degradation % 48 h			Degradation % 72 h		
		MRS Broth	Trypticase in Soy Broth	Nutrient Broth	MRS Broth	Trypticase in Soy Broth	Nutrient Broth	MRS Broth	Trypticase in Soy Broth	Nutrient Broth
1	FSL-1	39.3 ± 0.04	35.3 ± 0.01	32.1 ± 0.04	61.7 ± 0.02	58.8 ± 0.02	57.3 ± 0.02	75.7 ± 0.01	72.8 ± 0.02	74.9 ± 0.04
2	FSL-2	39.0 ± 0.01	32.8 ± 0.02	32.5 ± 0.02	61.0 ± 0.01	58.3 ± 0.04	58.0 ± 0.01	72.8 ± 0.01	70.9 ± 0.03	70.8 ± 0.02
3	FSL-5	36.5 ± 0.03	31.8 ± 0.03	31.0 ± 0.04	57.0 ± 0.03	53.0 ± 0.04	51.8 ± 0.03	70.6 ± 0.04	66.5 ± 0.01	67.0 ± 0.03
4	FSL-8	37.1 ± 0.02	33.8 ± 0.02	28.6 ± 0.01	62.0 ± 0.04	58.4 ± 0.02	56.2 ± 0.04	78.0 ± 0.02	72.5 ± 0.01	70.6 ± 0.01
5	FSL-17	38.6 ± 0.03	29.8 ± 0.02	27.9 ± 0.01	60.3 ± 0.02	56.3 ± 0.01	55.8 ± 0.03	72.6 ± 0.04	69.6 ± 0.04	69.0 ± 0.01
6	FSL-20	32.3 ± 0.02	30.8 ± 0.01	28.6 ± 0.01	51.2 ± 0.01	46.4 ± 0.01	48.2 ± 0.02	67.5 ± 0.03	61.8 ± 0.02	65.0 ± 0.03
7	FSL-21	31.1 ± 0.04	27.3 ± 0.03	26.7 ± 0.02	56.9 ± 0.04	52.1 ± 0.04	51.0 ± 0.01	69.5 ± 0.03	65.7 ± 0.03	65.5 ± 0.03
8	FSL-24	36.8 ± 0.03	28.5 ± 0.04	29.0 ± 0.04	60.7 ± 0.02	54.0 ± 0.03	53.3 ± 0.04	71.6 ± 0.01	66.7 ± 0.01	66.8 ± 0.01
9	FSL-27	31.7 ± 0.01	27.0 ± 0.01	26.4 ± 0.04	53.2 ± 0.03	48.0 ± 0.04	46.6 ± 0.03	72.0 ± 0.01	67.8 ± 0.01	67.3 ± 0.02
10	RS-5	34.3 ± 0.01	38.6 ± 0.02	33.6 ± 0.01	65.6 ± 0.04	59.3 ± 0.01	59.0 ± 0.03	78.3 ± 0.01	73.8 ± 0.04	72.8 ± 0.01

**Table 4 toxins-13-00712-t004:** Comparison of ZEA neutralization analysis result of potent bacterial strains through ELISA and HPLC.

Strain Codes	ELISA %	HPLC %
FSL-1	75.7	71.2
FSL-2	72.8	72.3
FSL-5	70.6	69.7
FSL-8	78.0	72.4
FSL-12	73.1	70.3
FSL-13	66.8	65.3
FSL-17	72.6	70.2
FSL-20	67.5	69.0
FSL-21	69.5	69.2
FSL-24	71.6	69.8
FSL-26	69.0	68.5
FSL-27	72.0	70.0
FSL-30	68.5	68.0
RS-5	78.3	74.3
RS-6	65.9	64.2
RS-13	64.4	64.0
RS-17	67.0	66.4
Average	70.77	69.11
Standard Deviation (±)	3.935	2.768
*p* value	*p* < 0.05	

**Table 5 toxins-13-00712-t005:** List of the compounds detected by GC-MS after ZEA degradation by RS-5.

S.No	Compound	Retention Time (RT)	Area %	Probability
1	1 propene,3,3ꞌ-oxy bis	0.43	40.56	21.01
2	4,8 Dioxaspiro[2.5] oct-1-ene 6,6 dimethyl	0.43	40.56	7.04
3	3-pentyn-1-ol	0.43	40.56	5.25
4	3,5,9-trioxa-5-phosphahepatacos-8-en-1-aminium	1.69	0.46	18.53
5	pyrrolo[1.2-q] pyrazine-1,4-dione,hexahydro	2.69	6.17	56.34
6	Glycyl-L-proline	2.69	6.17	30.75
7	Pyrrolo[1,2-a] pyrazine-1,4 dione, hexahydro-3-(2-methyl propyl)-	3.26	2.85	40.66
8	9-octadecenoic acid, (2-phenyl-1,3-dioxolan-4-y) methyl ester, cis	3.81	0.48	36.79
9	Hexasiloxane,1,1,3,3,5,5,7,7,9,9,11,11 dodecamethyl	5.61	8.21	35.62
10	1-monolinoleoylglycerol trymethylsilyl ether	7.47	0.12	53.03
11	1-monolinoleoylglycerol trymethylsilyl ether	8.05	0.25	54.44
12	Octasiloxane,1,1,3,3,5,5,7,7,9,9,11,11,13,13,15,15-hexadecamethyl-	9.57	40.27	41.45
13	Octasiloxane,1,1,3,3,5,5,7,7,9,9,11,11,13,13,15,15-hexadecamethyl-	10.51	0.26	40.70
14	Octasiloxane,1,1,3,3,5,5,7,7,9,9,11,11,13,13,15,15-hexadecamethyl-	10.94	0.24	50.87

## Data Availability

The data presented in this study are available on request from the corresponding author.

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
