# Peer review of "Biological Transformation of Zearalenone by Some Bacterial Isolates Associated with Ruminant and Food Samples"

_toxins, 2021, doi:10.3390/toxins13100712_

Round 1
Reviewer 1 Report
Comments and Suggestions for Authors
Zearalenone (ZEA) is one of the naturally occurring xenoestrogens that demonstrated genotoxic, teratogenic, hemotoxic, immunotoxic, carcinogenic, and hepatotoxic properties. It is produced by several Fusarium species presented in cereals such as barley, sorghum, oats, maize or wheat and can endanger human health via direct consumption of contaminated food as well as indirect consumption in contaminated animal products. Therefore, efficient methods for ZEA elimination/neutralization from foodstuff and fodder are in high demand.
In the submitted manuscript ‘Biological Transformation of Zearalenone by Some Bacterial Isolates Associated with Ruminant and Food Samples’ authors used a quite large number of food and rumen samples in order to isolate bacterial strains that demonstrated potential to neutralize ZEA. For identification of the isolated bacterial strains both morphological, cultural, biochemical as well as molecular techniques were used that enabled deep and reliable classification of the obtained isolates. Furthermore, the authors selected the most efficient strains in the ZEA neutralization using a multi-instrumental approach – ELISA tests, HPLC, and GC-MS analysis. In the result, the authors obtained ca 70% reduction of the ZEA concentration in the culture medium, which could be considered as considerable. All in all, the main findings of this manuscript are promising in view of the possibility to use the selected strains for ZEA neutralization in the future, however, I have a few comments that should be taken into account to improve manuscript quality and informative value for future readers.
- In my opinion the term ‘ZEA biodegradation’ should be changed into more universal ‘neutralization’. It is known that microbial neutralization of ZEA is mainly based on its biosorption by microorganism cells involving one or more of the three phenomena: (i) the physical binding of ZEA to surface components of the cell, (ii) the transport of ZEA inside the cell and then its accumulation, (iii) the metabolization of ZEA to a less toxic form. Using the term ‘degradation’ authors suggest that ZEA molecules are disrupted, however, measuring changes in the ZEA concentration in the medium does not allow to determinate exact mechanism responsible for the decrease of ZEA concentration. I suggest replace ‘biodegradation’ with ‘neutralization’ and add short description within the introduction section.
- Description of the molecular identification of the strains is too lapidary – it should contain information about PCR primers used for amplification of 16S rDNA, gradient program as well as PCR apparatus. Information about PCR primers which already are present in the results section should be placed in the M&M section. Moreover, a description of DNA sequencing should be also supplemented with information on whether sequencing was performed by authors or send to the external institution (provide name).
- In my opinion in the description of the HPLC analysis authors shouldn’t use the term “bio-transforming” since they only measured ZEA concentration – not their metabolites like α- and β-ZOL. For measurement of the α- and β-ZOL authors should use HPLC-ESI-MS/MS technique.
- Why for GC-MS analysis authors did not apply some kind of sample preparation methods like SPE or SPME that is frequently used by many authors to enriched the ZEA in the samples?
- Authors used culture media such as MRS broth, tryptic soy broth (TSB), and nutrient broth for investigation of the decrease of ZEA concentration. It is known that peptides and denatured proteins present in the culture media may participate in mycotoxin binding and therefore, probably can be responsible for binding the ZEA mycotoxin. Such phenomenon was noted in the work Złoch et al. 2020 (https://doi.org/10.1016/j.toxicon.2020.03.011). Please explain and discuss the possible impact of the mentioned phenomenon on the obtained results.
- Description of the results of the molecular identification of the investigated bacteria should be supplemented with the table that contains information about: percentage of the similarity of the investigated strain with the best match in NCBI database or EZbiocloude, the accession number from GenBank or at least the length of the obtained contig sequences (should be at least 1300 bp). Please also carefully revised the results of the molecular identification and compare with the accepted norms, that is, which percentage of the similarity is considered as sufficient to species or genus classification – it especially applied to results <97%. In my opinion, such low ID % is enough only for reliable genus classification – not species.
Author Response
We are thankful to editor and reviewers for timely completion of review process and providing us with valuable feedback.
Response to Reviewer # 1
Dear reviewer, we are grateful to you for your comments and suggestions for the improvement of our research manuscript. We have tried our best to revise the manuscript in light of your comments.
Comment: In my opinion the term ‘ZEA biodegradation’ should be changed into more universal ‘neutralization’. It is known that microbial neutralization of ZEA is mainly based on its biosorption by microorganism cells involving one or more of the three phenomena: (i) the physical binding of ZEA to surface components of the cell, (ii) the transport of ZEA inside the cell and then its accumulation, (iii) the metabolization of ZEA to a less toxic form. Using the term ‘degradation’ authors suggest that ZEA molecules are disrupted, however, measuring changes in the ZEA concentration in the medium does not allow to determinate exact mechanism responsible for the decrease of ZEA concentration. I suggest replace ‘biodegradation’ with ‘neutralization’ and add short description within the introduction section.
Reply: Corrected. See Line No. 56, 59-63, 65, 72, 110, 283, 284. Highlighted with green colour.
Comment: Description of the molecular identification of the strains is too lapidary – it should contain information about PCR primers used for amplification of 16S rDNA, gradient program as well as PCR apparatus. Information about PCR primers which already are present in the results section should be placed in the M&M section. Moreover, a description of DNA sequencing should be also supplemented with information on whether sequencing was performed by authors or send to the external institution (provide name).
Reply: Corrected. See Lines No. 106-109. Highlighted with green colour.
Comment: In my opinion in the description of the HPLC analysis authors shouldn’t use the term “bio-transforming” since they only measured ZEA concentration – not their metabolites like α- and β-ZOL. For measurement of the α- and β-ZOL authors should use HPLC-ESI-MS/MS technique.
Reply: Corrected. See Line No.283, 284, 285, 286. α- and β-ZOL were not measured as HPLC-ESI-MS/MS was not available. In GC-MS results these two compounds were not detected.
Comment: Why for GC-MS analysis authors did not apply some kind of sample preparation methods like SPE or SPME that is frequently used by many authors to enriched the ZEA in the samples?
Reply: McMaster et al., (2019) method was used by authors and it did not include enrichment procedure.
Comment: Authors used culture media such as MRS broth, tryptic soy broth (TSB), and nutrient broth for investigation of the decrease of ZEA concentration. It is known that peptides and denatured proteins present in the culture media may participate in mycotoxin binding and therefore, probably can be responsible for binding the ZEA mycotoxin. Such phenomenon was noted in the work Złoch et al. 2020 (https://doi.org/10.1016/j.toxicon.2020.03.011). Please explain and discuss the possible impact of the mentioned phenomenon on the obtained results.
Reply: Corrected. See Line No.351-354. The reference is added and discussed.
Comment: Description of the results of the molecular identification of the investigated bacteria should be supplemented with the table that contains information about: percentage of the similarity of the investigated strain with the best match in NCBI database or EZbiocloude, the accession number from GenBank or at least the length of the obtained contig sequences (should be at least 1300 bp). Please also carefully revised the results of the molecular identification and compare with the accepted norms, that is, which percentage of the similarity is considered as sufficient to species or genus classification – it especially applied to results <97%. In my opinion, such low ID % is enough only for reliable genus classification – not species.
Reply: Corrected.
Thanks for suggestions; the recommended table with desired information has been incorporated in the manuscript Line# 231,241. Accession numbers are added See Line No.220, 230, 231. Although, the sequence similarity of few samples was found below 98%, which is the best option for selection of novel specie, but the possibility of claiming novel specie or genera was excluded due to partial nucleotide sequence size which varies from 1152 to 1439. As we have reasonable information about nucleotide sequences similarities based taxonomy studies for claiming novel species and genera, therefore up to our knowledge due to partial size of the nucleotide sequences of our samples, we can only presume these samples as possible sub-species or strains of top hit bacterial isolates.
Reviewer 2 Report
The authors reported a finding of ZEA biotransformation by some bacterial isolates. Since many pervious works in this field have been published, the novel findings such as the identification of biotransformation product(s) would increase the significance of this work. The authors mentioned some possible byproducts of ZEA using GC-MS method. However, more clear experiment design and data analysis (e.g. the condition of GC-MS, the MS spectrums of the byproducts, and the details of MS analysis should be provided to support the conclusion.
Except for the main concern, there are some mistakes or typos listed below. Please also check the whole context carefully to make an correction.
Line 80: What is the concentration of methanol?
Line 116: Please use the unit of ng mL-1 instead of ppb.
Line 139: Please add the information of detector here.
Line 142: Please remove “(control sample)”.
Line 145-158: What is the MS condition?
Line 153: Please change “0f” to “of”.
Line 201: Please change “OF” to “of”.
Line 277-279: Did the author performed the replication (e.g. n=3) of this experiment? The conclusion must be made with the significant different between different media.
Line 286-288: Remove “The area of ZEA peak …”
Line 296-303: The authors carried out the T-test to compare two different methods (ELISA and HPLC). However, no conclusion was made in this result section.
Line 303-310: Please rewrite the section since the context, figure (10) and table (5) are confused. It is very hard to see any peaks related to the ZEA biotransformation because ZEA is almost 100% of relative abundance.
Line 314- : Please discuss more on the previous findings of ZEA biotransformation products and on the significance of this work.
The quality of Fig. 3 and 4 should be improved.
Author Response
We are thankful to editor and reviewers for timely completion of review process and providing us with valuable feedback.
Response to Reviewer # 2
Dear reviewer, we are grateful to you for your comments and suggestions for the improvement of our research manuscript. We have tried our best to revise the manuscript in light of your comments.
Comment: The authors reported a finding of ZEA biotransformation by some bacterial isolates. Since many pervious works in this field have been published, the novel findings such as the identification of biotransformation product(s) would increase the significance of this work. The authors mentioned some possible byproducts of ZEA using GC-MS method. However, more clear experiment design and data analysis (e.g. the condition of GC-MS, the MS spectrums of the byproducts, and the details of MS analysis should be provided to support the conclusion.
Reply: This study includes data of a research plan for a graduate study. In future, further research may be planned as suggested by the reviewer.
Comment: Except for the main concern, there are some mistakes or typos listed below. Please also check the whole context carefully to make an correction.
Comment: Line 80: What is the concentration of methanol?
Reply: Corrected. See Line No.85. Highlighted with green colour
Comment: Line 116: Please use the unit of ng mL-1 instead of ppb.
Reply: Corrected. See Line No 115, 124,125. Highlighted with green colour
Comment: Line 139: Please add the information of detector here.
Reply: Corrected. See Line No. 143,144. Highlighted with green colour
Comment: Line 142: Please remove “(control sample)”.
Reply: Corrected. See Line No.153. Highlighted with green colour
Comment: Line 145-158: What is the MS condition?
Reply: Corrected. See Line No164-171.Highlighted with green colour
Comment: Line 153: Please change “0f” to “of”.
Reply: Corrected. See Line No. 165. Highlighted with green colour
Comment: Line 201: Please change “OF” to “of”.
Reply: Corrected. See Line No. 211. Highlighted with green colour
Comment: Line 277-279: Did the author performed the replication (e.g. n=3) of this experiment? The conclusion must be made with the significant different between different media.
Reply: Corrected. See Line No.273. Highlighted with green colour
Comment: Line 286-288: Remove “The area of ZEA peak …”
Reply: Corrected. See Line No. 292. Highlighted with green colour
Comment: Line 296-303: The authors carried out the T-test to compare two different methods (ELISA and HPLC). However, no conclusion was made in this result section.
Reply: Corrected. See Line No.308, 309. Highlighted with green colour
Comment: Line 303-310: Please rewrite the section since the context, figure (10) and table (5) are confused. It is very hard to see any peaks related to the ZEA biotransformation because ZEA is almost 100% of relative abundance.
Reply: Corrected. See Line No.313-320. Highlighted with green colour
Comment: Line 314- : Please discuss more on the previous findings of ZEA biotransformation products and on the significance of this work.
Reply: Corrected. See Line No.351-354.Highlighted with green colour
Round 2
Reviewer 1 Report
I would like to thank the Authors for taking into account most of my comments, however, two important points remain unclear.
1) Previously I asked the Authors - Why for GC-MS analysis authors did not apply some kind of sample preparation methods like SPE or SPME that is frequently used by many authors to enriched the ZEA in the samples? - and Authors replied that McMaster et al., (2019) method was used in which authors did not include enrichment procedure. In mentioned work, authors (McMaster et al.) used sample preparation procedure before GC-MS analysis - please read "Sample Preparation and Extraction of Mycotoxins" section in that work, where You can find-
"Traditional mycotoxin extraction was performed using methods developed by Tacke and Casper (1996) and Mirocha et al. (1998). Eight milliliters of acetonitrile/water (86:14) were added to 1-g samples and then vortexed. The
samples were placed on an orbital shaker for 1 h at room temperature. Approximately 3 mL of the extraction solvent was passively filtered through a solid-phase extraction column containing 1 g of a mixture of C18 silica gel (60 Å, 40 to 63μm particle size range) and aluminum oxide (neutral, 50
to 200μm particle size range) (Sorbtech, Norcross, GA, USA) at a 1:3 ratio."
It means, as I suggested, that some sample preparation procedure before GC MS analysis should be done. If Authors cited McMaster it means that SPE method should be applied. Do I understand correctly? Please clarify this crucial issue.
2) I checked the accession numbers of the identified by Authors strains - two of them - D05 (Accession No.MZ452412) and F04 (Accession No MZ452410) have nucleotide sequence length far below 1300 bp considered as minimum length to secure species identification, therefore, Authors have no reason to claim that they represent P. gessardi species. In that cases, I suggest using the name Pseudomonas sp.
Author Response
October 02, 2021.
Dear Editor,
Greetings,
Thank you very much for your time and comments regarding our manuscript (toxins-1388253). Our manuscript “Biological Transformation of Zearalenone by Some Bacterial Isolates Associated With Ruminant and Food Samples” has been revised carefully and here we are giving our response to the reviewers’ comments. We have improved the manuscript according to the reviewers’ comments and suggestions. All the revisions can be easily identified from manuscript highlighted with Yellow color.
Once again thanks for your co-operation and valuable comments and suggestion. Moreover, the efforts of the reviewer are highly appreciated. We are hoping for pleasant response and further good comments (if any) from your side.
****************************************************************************************************
We are thankful to editor and reviewers for timely completion of review process and providing us with valuable feedback.
Response to Reviewer # 1
I would like to thank the Authors for taking into account most of my comments, however, two important points remain unclear.
Comment 1: Previously I asked the Authors - Why for GC-MS analysis authors did not apply some kind of sample preparation methods like SPE or SPME that is frequently used by many authors to enriched the ZEA in the samples? - and Authors replied that McMaster et al., (2019) method was used in which authors did not include enrichment procedure. In mentioned work, authors (McMaster et al.) used sample preparation procedure before GC-MS analysis - please read "Sample Preparation and Extraction of Mycotoxins" section in that work, where You can find-
"Traditional mycotoxin extraction was performed using methods developed by Tacke and Casper (1996) and Mirocha et al. (1998). Eight milliliters of acetonitrile/water (86:14) were added to 1-g samples and then vortexed. The
samples were placed on an orbital shaker for 1 h at room temperature. Approximately 3 mL of the extraction solvent was passively filtered through a solid-phase extraction column containing 1 g of a mixture of C18 silica gel (60 Å, 40 to 63μm particle size range) and aluminum oxide (neutral, 50
to 200μm particle size range) (Sorbtech, Norcross, GA, USA) at a 1:3 ratio."
It means, as I suggested, that some sample preparation procedure before GC MS analysis should be done. If Authors cited McMaster it means that SPE method should be applied. Do I understand correctly? Please clarify this crucial issue.
Reply: Corrected. See Line No. 157, 158, 165-174. 29 No. reference is also added, Replaced with previous one. Highlighted with yellow colour.
Comment: I checked the accession numbers of the identified by Authors strains - two of them - D05 (Accession No.MZ452412) and F04 (Accession No MZ452410) have nucleotide sequence length far below 1300 bp considered as minimum length to secure species identification, therefore, Authors have no reason to claim that they represent P. gessardi species. In that cases, I suggest using the name Pseudomonas sp.
Reply: Corrected. See Line No.244-249.251-252.
